# Learning Hierarchical Structures with Differentiable Nondeterministic Stacks

**Brian DuSell and David Chiang**
Department of Computer Science and Engineering, University of Notre Dame
{bdusell1,dchiang}@nd.edu

## Abstract

Learning hierarchical structures in sequential data—from simple algorithmic patterns to natural language—in a reliable, generalizable way remains a challenging problem for neural language models. Past work has shown that recurrent neural networks (RNNs) struggle to generalize on held-out algorithmic or syntactic patterns without supervision or some inductive bias. To remedy this, many papers have explored augmenting RNNs with various differentiable stacks, by analogy with finite automata and pushdown automata (PDAs). In this paper, we improve the performance of our recently proposed Nondeterministic Stack RNN (NS-RNN), which uses a differentiable data structure that simulates a nondeterministic PDA, with two important changes. First, the model now assigns unnormalized positive weights instead of probabilities to stack actions, and we provide an analysis of why this improves training. Second, the model can directly observe the state of the underlying PDA. Our model achieves lower cross-entropy than all previous stack RNNs on five context-free language modeling tasks (within 0.05 nats of the information-theoretic lower bound), including a task on which the NS-RNN previously failed to outperform a deterministic stack RNN baseline. Finally, we propose a restricted version of the NS-RNN that incrementally processes infinitely long sequences, and we present language modeling results on the Penn Treebank.

## 1 Introduction

Many machine learning problems involve sequential data with hierarchical structures, such as modeling context-free languages (Grefenstette et al., 2015; DuSell & Chiang, 2020), evaluating mathematical expressions (Nangia & Bowman, 2018; Hao et al., 2018), logical inference (Bowman et al., 2015), and modeling syntax in natural language (Dyer et al., 2016; Shen et al., 2019b; Kim et al., 2019). However, recurrent neural networks (RNNs) have difficulty learning to solve these tasks, or generalizing to held-out sequences, unless they have supervision or a hierarchical inductive bias (van Schijndel et al., 2019; Wilcox et al., 2019; McCoy et al., 2020). A limiting factor of RNNs is their reliance on memory whose size is constant across time. For example, to predict the second half of a string of the form $w\#w^{\mathrm{R}}$, a simple RNN would need to store all of $w$ in its hidden state before predicting its reversal $w^{\mathrm{R}}$; a memory of finite size will inevitably fail to do this for inputs exceeding a certain length.

To remedy this, some previous work has investigated the addition of differentiable stack data structures to RNNs (Sun et al., 1995; Grefenstette et al., 2015; Joulin & Mikolov, 2015; DuSell & Chiang, 2020), which is closely related to work on neural networks that model shift-reduce parsers (Bowman et al., 2016; Dyer et al., 2016; Shen et al., 2019a). Just as adding a stack to a finite state machine, which makes it a pushdown automaton (PDA), enables it to recognize context-free languages (CFLs), the hope is that adding stacks to RNNs will increase the range of problems on which they can be used effectively. We also expect stacks to aid training by introducing an inductive bias for learning hierarchical patterns, and to increase generalization power by structuring the model's memory in a way that better predicts held-out hierarchical data.

Previously (DuSell & Chiang, 2020), we proposed a stack-based RNN called the Nondeterministic Stack RNN (NS-RNN) that outperformed other stack RNNs on a range of CFL language modeling tasks. Its defining feature is that its external data structure is a nondeterministic PDA, allowing it to simulate an exponential number of sequences of stack operations in parallel. This is in contrast to prior

stack RNNs (Grefenstette et al., 2015; Joulin & Mikolov, 2015) which model deterministic stacks, being designed to learn *one* correct stack operation at each time step. One reason nondeterminism is important is that deterministic CFLs are a proper subset of CFLs. If the analogy with PDAs holds true, then equipping an RNN with a deterministic stack would only enable it to model deterministic CFLs, whereas a nondeterministic stack should enable it to model all CFLs. This is important for natural language processing, as human language is known to be high in syntactic ambiguity.

Another benefit of nondeterminism, even on deterministic CFLs, applies to training. In order for a model to receive a reward for an action, it must try the action (that is, give it nonzero probability so that it receives gradient during backpropagation). For example, in the digit-recognition task, a classifier tries all ten digits, and is rewarded for the correct one. But in a stack-augmented model, the space of possible action sequences is very large. Whereas a deterministic stack can only try one of them, a nondeterministic stack can try all of them and always receives a reward for the correct one. But as explained in §3.1, because the NS-RNN's probability for an action sequence is the product of many probabilities, it can be extremely small, so the NS-RNN sometimes learns very slowly.

In this paper we present a new model, the Renormalizing NS-RNN (RNS-RNN), which is based on the NS-RNN, but improves its performance on all of the CFL tasks it was originally tested on, thanks to two key changes. The first is that stack actions have weights that do not necessarily form a probability distribution (§3.1). They define an unnormalized distribution over stacks that is *renormalized* whenever the model queries it. Second, the RNS-RNN includes not only top stack symbols but also PDA states in this query (§3.2). These changes allow the RNS-RNN to attain lower cross-entropy on CFL tasks (in fact, very close to the information-theoretic lower bound) and to surpass deterministic stack RNNs on a task on which the NS-RNN fails to do so ("padded reversal"). Finally, as a third modification, we present a memory-restricted version of the RNS-RNN that requires only $O(n)$ time and space (§5). This restricted RNS-RNN can be run incrementally on arbitrarily long sequences, which is a necessity for language modeling on natural language, for which we provide experimental results. Our code is available at `https://github.com/bdusell/nondeterministic-stack-rnn`.

## 2 Previous stack RNNs

We begin by discussing three previously proposed stack RNNs, each of which uses a different style of differentiable stack: stratification (Das et al., 1992; Sun et al., 1995; Grefenstette et al., 2015), superposition (Joulin & Mikolov, 2015), and nondeterminism (DuSell & Chiang, 2020).

### 2.1 Controller-stack interface

Each type of stack RNN consists of a simple RNN (or variant such as LSTM), called the *controller*, connected to a differentiable stack. The stack has no parameters of its own; its role is to accept *actions* from the controller to push and pop elements at each time step, simulate those actions, and return a *reading* to the controller as an extra input to the next time step that serves as a representation of the updated top element of the stack. The stack actions and stack reading take on continuous values so that they may be differentiable; their form and interpretation vary with architecture.

Following prior work (DuSell & Chiang, 2020), we make minor changes to the original model definitions given by Grefenstette et al. (2015) and Joulin & Mikolov (2015) to ensure that all three of these stack RNN models conform to the same controller-stack interface. This allows us to isolate differences in the style of stack data structure employed while keeping other parts of the network the same. We assume the input $w = w_1 \cdots w_n$ is encoded as a sequence of vectors $\mathbf{x}_1, \cdots, \mathbf{x}_n$. In all of our experiments, we use an LSTM (Hochreiter & Schmidhuber, 1997) as the controller, whose memory consists of a hidden state $\mathbf{h}_t$ and memory cell $\mathbf{c}_t$. The controller computes the next state $(\mathbf{h}_t, \mathbf{c}_t)$ given the previous state $(\mathbf{h}_{t-1}, \mathbf{c}_{t-1})$, input vector $\mathbf{x}_t$, and stack reading $\mathbf{r}_{t-1}$:

$$(\mathbf{h}_t, \mathbf{c}_t) = \text{LSTM}\left((\mathbf{h}_{t-1}, \mathbf{c}_{t-1}), \begin{bmatrix} \mathbf{x}_t \\ \mathbf{r}_{t-1} \end{bmatrix}\right).$$

We set $\mathbf{h}_0 = \mathbf{c}_0 = \mathbf{0}$. The hidden state is used to compute the stack actions $a_t$ and predict the logits $\mathbf{y}_t$ for the next word $w_{t+1}$. The previous stack and new actions are used to compute a new stack $s_t$, which in turn is used to produce a new reading $\mathbf{r}_t$:

$$a_t = \text{Actions}(\mathbf{h}_t) \qquad \mathbf{y}_t = \mathbf{W}_{\text{hy}}\mathbf{h}_t + \mathbf{b}_{\text{hy}} \qquad s_t = \text{Stack}(s_{t-1}, a_{t-1}) \qquad \mathbf{r}_t = \text{Reading}(s_t)$$

In order to change the stack data structure, we need only change the definitions of ACTIONS, STACK, READING, and $s_0$, which may depend on parameters of the model; for our changes to the NS-RNN, we will only need to change ACTIONS and READING.

## 2.2 STRATIFICATION

Based on work by Das et al. (1992) and Sun et al. (1995), the stack of Grefenstette et al. (2015) relies on a strategy we have dubbed "stratification" (DuSell & Chiang, 2020). The elements of the stack are vectors, each of which is associated with a "thickness" between 0 and 1, which represents the degree to which the vector element is present on the stack. A helpful analogy is that of layers of a cake; the stack elements are like cake layers of varying thickness. In this model, $a_t = (u_t, d_t, \mathbf{v}_t)$, where the pop signal $u_t \in (0, 1)$ indicates the amount to be removed from the top of the stack, $\mathbf{v}_t$ is a learned vector to be pushed as a new element onto the stack, and the push signal $d_t \in (0, 1)$ is the thickness of that newly pushed vector. This model has quadratic time and space complexity with respect to input length. We refer the reader to Appendix A.1 for full details.

## 2.3 SUPERPOSITION

The stack of Joulin & Mikolov (2015) simulates a combination of partial stack actions by computing three new, separate stacks: one with all cells shifted down (push), kept the same (no-op), and shifted up (pop). The new stack is then an element-wise interpolation ("superposition") of these three stacks. In this model, stack elements are again vectors, and $a_t = (\mathbf{a}_t, \mathbf{v}_t)$, where the vector $\mathbf{a}_t$ is a probability distribution over three stack operations: push a new vector, no-op, and pop the top vector; $\mathbf{v}_t$ is the vector to be pushed. The vector $\mathbf{v}_t$ can be learned or can be set to $\mathbf{h}_t$ (Yogatama et al., 2018). The stack reading is the top cell of the stack. This model has quadratic time and space complexity with respect to input length. We refer the reader to Appendix A.2 for full details.

## 2.4 NONDETERMINISM

The stack module in the Nondeterministic Stack RNN (NS-RNN) model (DuSell & Chiang, 2020) maintains a probability distribution over whole stacks by simulating a weighted PDA. It has cubic time complexity and quadratic space complexity with respect to input length, leading to higher wall-clock run time than other stack RNNs, but often better task performance.

The simulated weighted PDA maintains a state drawn from a finite set $Q$, which includes an initial state $q_0$, and a stack with symbols drawn from an alphabet $\Gamma$, which includes an initial symbol $\perp$. At each time step, the PDA executes a weighted *transition* that changes its state and manipulates the stack. Stack operations are drawn from the set $\text{Op}(\Gamma) = \bullet\Gamma \cup \Gamma \cup \{\epsilon\}$, where for any $y \in \Gamma$, $\bullet y$ means "push $y$," $y$ means "replace top element with $y$," and $\epsilon$ means "pop top element." A valid sequence of transitions is called a *run*, and the weight of a run is the product of the weights of its transitions.

The RNN controller emits transition weights to the stack module. Note that the stack module, not the controller, keeps track of PDA states and stack configurations, so the controller emits *distributions* over transitions conditioned on the PDA's current state and top stack symbol. More precisely, $a_t = \Delta[t]$ is a tensor where the meaning of element $\Delta[t][q, x \to r, \upsilon]$ is: if the PDA is in state $q$ and the top stack symbol is $x$, then, with weight $\Delta[t][q, x \to r, \upsilon]$, go to state $r$ and perform stack operation $\upsilon \in \text{Op}(\Gamma)$. The original NS-RNN definition requires that for all $t$, $q$, and $x$, the weights form a probability distribution. Accordingly, they are computed from the hidden state using a softmax layer:

$$\Delta[t] = \underset{q,x}{\text{softmax}}(\mathbf{W}_{\text{ha}}\mathbf{h}_t + \mathbf{b}_{\text{ha}}). \tag{1}$$

The stack module marginalizes over all runs ending at time step $t$ and returns the distribution over top stack symbols at $t$ to the controller. It may appear that computing this distribution is intractable because the number of possible runs is exponential in $t$, but Lang (1974) gives a dynamic programming algorithm that simulates all runs of a nondeterministic PDA in cubic time and quadratic space. Lang's algorithm exploits structural similarities in PDA runs. First, multiple runs can result in the same stack. Second, for $k > 0$, a stack of height $k$ must have been derived from a stack of height $k - 1$, so in principle representing a stack of height $k$ requires only storing its top symbol and a pointer to a stack of height $k - 1$. The resulting data structure is a weighted graph where edges represent individual stack symbols, and paths (of which there are exponentially many) represent stacks.

We may equivalently view this graph as a weighted finite automaton (WFA) that encodes a distribution over stacks, and accordingly the NS-RNN's stack module is called the *stack WFA*. Indeed, the language of stacks at a given time step $t$ is always regular (Autebert et al., 1997), and Lang's algorithm gives an explicit construction for the WFA encoding this language. Its states are PDA configurations of the form $(i, q, x)$, where $0 \leq i \leq n$, $q \in Q$, and $x \in \Gamma$ is the stack top. A stack WFA transition from $(i, q, x)$ to $(t, r, y)$ means the PDA went from configuration $(i, q, x)$ to $(t, r, y)$ (possibly via multiple time steps), where the only difference in the stack is that a single $y$ was pushed, and the $x$ was never modified in between. The weights of these transitions are stored in a tensor $\gamma$ of shape $n \times n \times |Q| \times |\Gamma| \times |Q| \times |\Gamma|$, where elements are written $\gamma[i \to t][q, x \to r, y]$. For $0 \leq i < t \leq n$,

$$
\begin{aligned}
\gamma[i \to t][q, x \to r, y] = & \\
& \mathbb{I}[i = t{-}1] \, \Delta[t][q, x \to r, \bullet y] & \text{push} \\
& + \sum_{s,z} \gamma[i \to t{-}1][q, x \to s, z] \, \Delta[t][s, z \to r, y] & \text{repl.} \\
& + \sum_{k=i+1}^{t-2} \sum_u \sum_{s,z} \gamma[i \to k][q, x \to u, y] \, \gamma[k \to t{-}1][u, y \to s, z] \, \Delta[t][s, z \to r, \epsilon] & \text{pop}
\end{aligned}
\tag{2}
$$

The NS-RNN sums over all stacks (accepting paths in the stack WFA) using a tensor $\alpha$ of *forward weights* of shape $n \times |Q| \times |\Gamma|$. The weight $\alpha[t][r, y]$ is the total weight of reaching configuration $(t, r, y)$. These weights are normalized to get the distribution over top stack symbols at $t$:

$$
\alpha[0][r, y] = \mathbb{I}[r = q_0 \wedge y = \bot]
\tag{3}
$$

$$
\alpha[t][r, y] = \sum_{i=1}^{t-1} \sum_{q,x} \alpha[i][q, x] \, \gamma[i \to t][q, x \to r, y] \qquad (1 \leq t \leq n)
\tag{4}
$$

$$
\mathbf{r}_t[y] = \frac{\sum_r \alpha[t][r, y]}{\sum_{y'} \sum_r \alpha[t][r, y']}.
\tag{5}
$$

We refer the reader to our earlier paper (DuSell & Chiang, 2020) for details of deriving these equations from Lang's algorithm. To avoid underflow and overflow, in practice, $\Delta$, $\gamma$, and $\alpha$ are computed in log space. The model's time complexity is $O(|Q|^4|\Gamma|^3 n^3)$, and its space complexity is $O(|Q|^2|\Gamma|^2 n^2)$.

## 3 Renormalizing NS-RNN

Here, we introduce the Renormalizing NS-RNN, which differs from the NS-RNN in two ways.

### 3.1 Unnormalized transition weights

To make a good prediction at time $t$, the model may need a certain top stack symbol $y$, which may in turn require previous actions to be orchestrated correctly. For example, consider the language $\{v\#v^{\mathrm{R}}\}$, where $n$ is odd and $w_t = w_{n-t+1}$ for all $t$. In order to do better than chance when predicting $w_t$ (for $t$ in the second half), the model has to push a stack symbol that encodes $w_t$ at time $(n - t + 1)$, and that same symbol must be on top at time $t$. How does the model learn to do this? Assume that the gradient of the log-likelihood with respect to $\mathbf{r}_t[y]$ is positive; this gradient "flows" to the PDA transition probabilities via (among other things) the partial derivatives of $\log \alpha$ with respect to $\log \Delta$.

To calculate these derivatives more easily, we express $\alpha$ directly (albeit less efficiently) in terms of $\Delta$:

$$
\alpha[t][r, y] = \sum_{\delta_1 \cdots \delta_t \rightsquigarrow r, y} \prod_{i=1,\ldots,t} \Delta[i][\delta_i]
$$

where each $\delta_i$ is a PDA transition of the form $q_1, x_1 \to q_2, x_2$, $\Delta[i][\delta_i] = \Delta[i][q_1, x_1 \to q_2, x_2]$, and the summation over $\delta_1 \cdots \delta_t \rightsquigarrow r, y$ means that after following transitions $\delta_1, \ldots, \delta_t$, then the PDA will be in state $r$ and its top stack symbol will be $y$. Then the partial derivatives are:

$$
\frac{\partial \log \alpha[t][r, y]}{\partial \log \Delta[i][\delta]} = \frac{\sum_{\delta_1 \cdots \delta_t \rightsquigarrow r, y} \left( \prod_{i'=1}^{t} \Delta[i'][\delta_{i'}] \right) \mathbb{I}[\delta_i = \delta]}{\sum_{\delta_1 \cdots \delta_t \rightsquigarrow r, y} \prod_{i'=1}^{t} \Delta[i][\delta_{i'}]}.
$$

This is the posterior probability of having used transition $\delta$ at time $i$, given that the PDA has read the input up to time $t$ and reached state $r$ and top stack symbol $y$.

So if a correct prediction at time $t$ depends on a stack action at an earlier time $i$, the gradient flow to that action is proportional to its probability given the correct prediction. This probability is always nonzero, as desired. However, this probability is the product of individual action probabilities, which are always strictly less than one. If a correct prediction depends on orchestrating many stack actions, then this probability may become very small. Returning to our example, we expect the model to begin by learning to predict the middle of the string, where only a few stack actions must be orchestrated, then working its way outwards, more and more slowly as more and more actions must be orchestrated. In §4 we verify empirically that this is the case.

The solution we propose is to use unnormalized (non-negative) transition weights, not probabilities, and to normalize weights only when reading. Equation (1) now becomes

$$\Delta[t] = \exp(\mathbf{W}_{\text{ha}}\mathbf{h}_t + \mathbf{b}_{\text{ha}}).$$

The gradient flowing to a transition is still proportional to its posterior probability, but now each transition weight has the ability to "amplify" (Lafferty et al., 2001) other transitions in shared runs. Equation (5) is not changed (yet), but its interpretation is. The NS-RNN maintains a probability distribution over stacks and updates it by performing probabilistic operations. Now, the model maintains an unnormalized weight distribution, and when it reads from the stack at each time step, it renormalizes this distribution and marginalizes it to get a probability distribution over readings. For this reason, we call our new model a Renormalizing NS-RNN (RNS-RNN).

### 3.2 PDA STATES INCLUDED IN STACK READING

In the NS-RNN, the controller can read the distribution over the PDA's current top stack symbol, but it cannot observe its current state. To see why this is a problem, consider the language $\{vv^{\text{R}}\}$. While reading $v$, the controller should predict the uniform distribution, but while reading $v^{\text{R}}$, it should predict based on the top stack symbol. A PDA with two states can nondeterministically guess whether the current position is in $v$ or $v^{\text{R}}$. The controller should interpolate the two distributions based on the weight of being in each state, but it cannot do this without input from the stack WFA, since the state is entangled with the stack contents. We solve this in the RNS-RNN by computing a joint distribution over top stack symbols *and* PDA states, making $\mathbf{r}_t$ a vector of size $|Q||\Gamma|$. Equation 5 becomes

$$\mathbf{r}_t[(r,y)] = \frac{\alpha[t][r,y]}{\sum_{r',y'} \alpha[t][r',y']}.$$

## 4 EXPERIMENTS ON FORMAL LANGUAGES

In order to assess the benefits of using unnormalized transition weights and including PDA states in the stack reading, we ran the RNS-RNN with and without the two proposed modifications on the same five CFL language modeling tasks used previously (DuSell & Chiang, 2020). We use the same experimental setup and PCFG settings, except for one important difference: we require the model to predict an end-of-sequence (EOS) symbol at the end of every string. This way, the model defines a proper probability distribution over strings, improving the interpretability of the results.

Each task is a weighted CFL specified as a PCFG:

**Marked reversal** The palindrome language with a middle marker ($\{v\#v^{\text{R}} \mid v \in \{\mathbf{0},\mathbf{1}\}^*\}$).

**Unmarked reversal** The palindrome language without a middle marker ($\{vv^{\text{R}} \mid v \in \{\mathbf{0},\mathbf{1}\}^*\}$).

**Padded reversal** Like unmarked reversal, but with a long stretch of repeated symbols in the middle ($\{va^p v^{\text{R}} \mid v \in \{\mathbf{0},\mathbf{1}\}^*, a \in \{\mathbf{0},\mathbf{1}\}, p \geq 0\}$).

**Dyck language** The language $D_2$ of strings with balanced brackets (two bracket types).

**Hardest CFL** A language shown by Greibach (1973) to be at least as hard to parse as any other CFL.

The marked reversal and Dyck languages are deterministic tasks that could be solved optimally with a deterministic PDA. On the other hand, the unmarked reversal, padded reversal, and hardest CFL tasks

require nondeterminism, with hardest CFL requiring the most (DuSell & Chiang, 2020, Appendix A). We randomly sample from these languages to create training, validation, and test sets. All strings are represented as sequences of one-hot vectors. Please see Appendix B for additional experimental details.

We evaluate models according to per-symbol cross-entropy (lower is better). For any set of strings $S$ and probability distribution $p$, it is defined as

$$H(S, p) = \frac{-\sum_{w \in S} \log p(w \cdot \text{EOS})}{\sum_{w \in S}(|w| + 1)}.$$

Since the validation and test strings are all sampled from known distributions, we can also use this formula to compute the per-symbol entropy of the true distribution (DuSell & Chiang, 2020). In our experiments we measure performance as the difference between the model cross-entropy and the true entropy, per-symbol and measured in nats (lower is better, and zero is optimal).

We compare seven models on the CFL tasks, each of which consists of an LSTM connected to a different type of stack: none ("LSTM"); stratification ("Gref"); superposition ("JM"); nondeterministic, aka NS-RNN ("NS"); NS with PDA states in the reading and normalized action weights ("NS+S"); NS with no states in the reading and unnormalized action weights ("NS+U"); and NS with PDA states and unnormalized action weights, or RNS-RNN ("NS+S+U").

**Results** We show validation set performance as a function of training time in Figure 1, and test performance binned by string length in Figure 2 (see also Appendix C for wall-clock training times). For all tasks, we see that our RNS-RNN (denoted NS+S+U) attains near-optimal cross-entropy (within 0.05 nats) on the validation set. All stack models effectively solve the deterministic marked reversal and Dyck tasks, although we note that on marked reversal the NS models do not generalize well on held-out lengths. Our new model excels on the three nondeterministic tasks: unmarked reversal, padded reversal, and hardest CFL. We find that the combination of both enhancements (+S+U) greatly improves performance on unmarked reversal and hardest CFL over previous work. For unmarked reversal, merely changing the task by adding EOS causes the baseline NS model to perform worse than Gref and JM; this may be because it requires the NS-RNN to learn a correlation between the two most distant time steps. Both enhancements (+S+U) in the RNS-RNN are essential here; without unnormalized weights, the model does not find a good solution during training, and without PDA states, the model does not have enough information to make optimal decisions. For padded reversal, we see that the addition of PDA states in the stack reading (+S) proves essential to improving performance. Although NS+S and NS+S+U have comparable performance on padded reversal, NS+S+U converges much faster. On hardest CFL, using unnormalized weights by itself (+U) improves performance, but only both modifications together (+S+U) achieve the best performance.

In Figure 3, we show the evolution of stack actions for the NS+S (normalized) and NS+S+U (unnormalized) models over training time on the simplest of the CFL tasks: marked reversal. We see that the normalized model begins solving the task by learning to push and pop symbols close to the middle marker. It then gradually learns to push and pop matching pairs of symbols further and further away from the middle marker. On the other hand, the unnormalized model learns the correct actions for all time steps almost immediately.

## 5 Incremental execution

Having demonstrated improvements on synthetic tasks, we now turn to language modeling on natural language. For standard language modeling benchmarks, during both training and evaluation, RNN language models customarily process the entire data set in order as if it were one long sequence, since being able to retain contextual knowledge of past sentences significantly improves predictions for future sentences. Running a full forward and backward pass during training on such a long sequence would be infeasible, so the data set is processed incrementally using a technique called truncated backpropagation through time (BPTT). This technique is feasible for models whose time and space complexity is linear with respect to sequence length, but for memory-augmented models such as stack RNNs, something must be done to limit the time and storage requirements. Yogatama et al. (2018) did this for the superposition stack by limiting the stack to 10 elements. In this section, we propose a technique for limiting the space and time requirements of the RNS-RNN (or NS-RNN), allowing us to use truncated BPTT and retain contextual information.

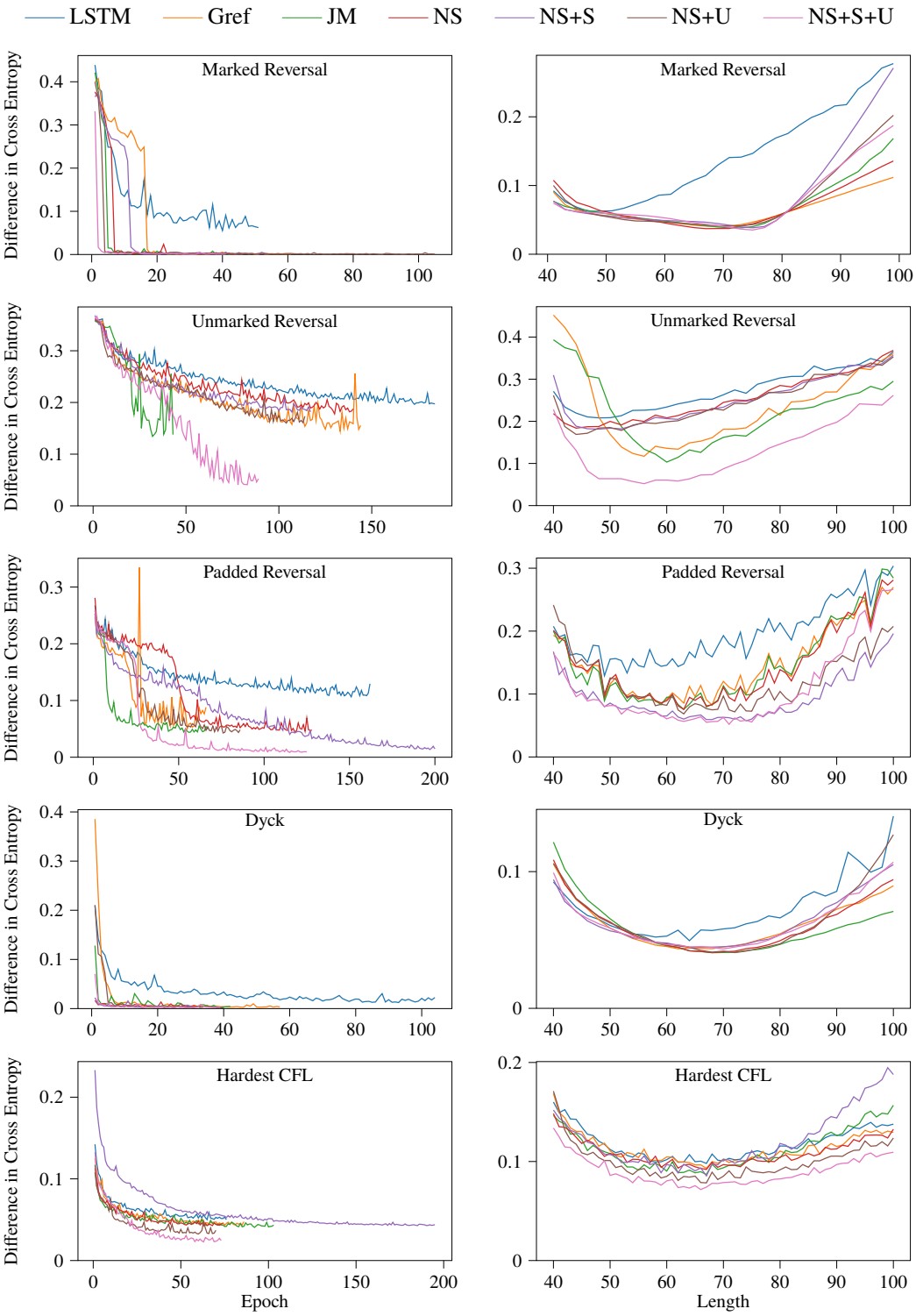

Figure 1: Cross-entropy difference in nats between model and source distribution on validation set vs. training time. Each line corresponds to the model which attains the lowest difference in cross-entropy out of all random restarts.

Figure 2: Cross-entropy difference in nats on the test set, binned by string length. These models are the same as those shown in Figure 1.

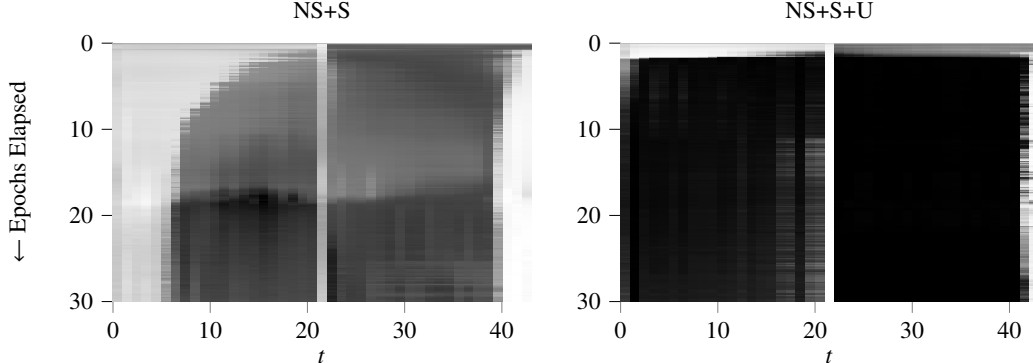

Figure 3: Visualization of the first 30 epochs of training (top to bottom) on the marked reversal task. In each plot, the horizontal axis is the string position (time step). Darkness indicates the weight assigned to the correct stack action type, normalized by the weight of all actions at time $t$ (black = correct, white = incorrect). The white band in the middle occurs because "replace" is considered the correct action type for the middle time step, but the models apparently learned to perform a different action without affecting the results. Both models were trained with learning rate 0.005.

## 5.1 Memory-limited RNS-RNN

We introduce the constraint that the stack WFA can only contain transitions $\gamma[i \to t][q, x \to r, y]$ where $t - i$ does not exceed a hyperparameter $D$; all other transitions are treated as having zero weight. It may be easier to get an intuition for this constraint in terms of CFGs. The stack WFA formulation is based on Lang's algorithm (1974), which can be thought of as converting a PDA to a CFG and then parsing with a CKY-style algorithm. The equation for $\gamma$ (Equation 2) has three terms, corresponding to rules of the form $(A \to b)$ (push), $(A \to Bc)$ (replace), and $(A \to BCd)$ (pop). The constraint $t - i \leq D$ on $\gamma$ means that these rules can only be used when they span at most $D$ positions.

The equations for $\alpha$ (3–4) have two cases, which correspond to rules of the form $(A \to \epsilon)$ and $(A \to AB)$. The definition of $\alpha$ allows these rules to be used for spans starting at 0 and ending anywhere. This is essentially equivalent to the constraint used in the Hiero machine translation system (Chiang, 2005), which uses synchronous CFGs under the constraint that no nonterminal spans more than 10 symbols, with the exception of so-called glue rules $S \to X$, $S \to SX$.

As a consequence of this constraint, if we consider the tensor $\gamma$, which contains the weights of the stack WFA, as a matrix with axes for the variables $i$ and $t$, then the only non-zero entries in $\gamma$ lie in a band of height $D$ along the diagonal. Crucially, column $t$ of $\gamma$ depends only on $\gamma[i \to t']$ for $t - D \leq i \leq t - 2$ and $t - D + 1 \leq t' \leq t - 1$. Similarly, $\alpha[t]$ depends only on $\alpha[i]$ for $t - D \leq i \leq t - 1$ and $\gamma[i \to t]$ for $t - i \leq D$. So, just as truncated BPTT for an RNN involves freezing the hidden state and forwarding it to the next forward-backward pass, truncated BPTT for the (R)NS-RNN involves forwarding the hidden state of the controller *and* forwarding a slice of $\gamma$ and $\alpha$. This reduces the time complexity of the (R)NS-RNN to $O(|Q|^4|\Gamma|^3D^2n)$ and its space complexity to $O(|Q|^2|\Gamma|^2Dn)$.

## 5.2 Experiments

Limiting the memory of the (R)NS-RNN now makes it feasbile to run experiments on natural language modeling benchmarks, although the high computational cost of increasing $|Q|$ and $|\Gamma|$ still limits us to settings with little information bandwidth in the stack. We believe this will make it difficult for the (R)NS-RNN to store lexical information on the stack, but it might succeed in using $\Gamma$ as a small set of syntactic categories. To this end, we ran exploratory experiments with the NS-RNN, RNS-RNN, and other language models on the Penn Treebank (PTB) as preprocessed by Mikolov et al. (2011).

We compare four types of model: LSTM, superposition ("JM") with a maximum stack depth of 10, and memory-limited NS-RNNs ("NS") and RNS-RNNs ("RNS") with $D = 35$. We based the hyperparameters for our LSTM baseline and training schedule on those of Semeniuta et al. (2016) (details in Appendix D). We also test two variants of JM, pushing either the hidden state or a learned

Table 1: Language modeling results on PTB, measured by perplexity and SG score. The setting $|Q| = 1$, $|\Gamma| = 2$ represents minimal capacity in the (R)NS-RNN models and is meant to serve as a baseline for the other settings. The other two settings are meant to test the upper limits of model capacity before computational cost becomes too great. The setting $|Q| = 1$, $|\Gamma| = 11$ represents the greatest number of stack symbol types we can afford to use, using only one PDA state. We selected the setting $|Q| = 3$, $|\Gamma| = 4$ by increasing the number of PDA states, and then the number of stack symbol types, until computational cost became too great (recall that the time complexity is $O(|Q|^4|\Gamma|^3)$, so adding states is more expensive than adding stack symbol types).

| Model | # Params | Val | Test | SG Score |
|---|---|---|---|---|
| LSTM, 256 units | 5,656,336 | 125.78 | 120.95 | 0.433 |
| LSTM, 258 units | 5,704,576 | 122.08 | 118.20 | 0.420 |
| LSTM, 267 units | 5,922,448 | 125.20 | 120.22 | 0.437 |
| JM (push hidden state), 247 units | 5,684,828 | **121.24** | **115.35** | 0.387 |
| JM (push learned), $|\mathbf{v}_t| = 22$ | 5,685,289 | 122.87 | 117.93 | 0.431 |
| NS, $|Q| = 1$, $|\Gamma| = 2$ | 5,660,954 | 126.10 | 122.62 | 0.414 |
| NS, $|Q| = 1$, $|\Gamma| = 11$ | 5,732,621 | 129.11 | 124.98 | 0.431 |
| NS, $|Q| = 3$, $|\Gamma| = 4$ | 5,743,700 | 126.71 | 122.53 | 0.447 |
| RNS, $|Q| = 1$, $|\Gamma| = 2$ | 5,660,954 | 122.64 | 117.56 | 0.435 |
| RNS, $|Q| = 1$, $|\Gamma| = 11$ | 5,732,621 | 127.21 | 121.84 | 0.386 |
| RNS, $|Q| = 3$, $|\Gamma| = 4$ | 5,751,892 | 122.67 | 118.09 | 0.408 |

vector. Unless otherwise noted, the LSTM controller has 256 units, one layer, and no dropout. For each model, we randomly search for initial learning rate and gradient clipping threshold; we report results for the model with the best validation perplexity out of 10 random restarts. In addition to perplexity, we also report the recently proposed Syntactic Generalization (SG) score metric (Hu et al., 2020; Gauthier et al., 2020). This score, which ranges from 0 to 1, puts a language model through a battery of psycholinguistically-motivated tests that test how well a model generalizes to non-linear, nested syntactic patterns. Hu et al. (2020) noted that perplexity does not, in general, agree with SG score, so we hypothesized the SG score would provide crucial insight into the stack's effectiveness.

**Results**   We show the results of our experiments on the Penn Treebank in Table 1. We reproduce the finding of Yogatama et al. (2018) that JM can achieve lower perplexity than an LSTM with a comparable number of parameters, but this does not translate into a better SG score. The results for NS and RNS do not show a clear trend in perplexity or SG score as the number of states or stack symbols increases, or as the modifications in RNS are applied, even when breaking down SG score by type of syntactic test (see Appendix E). We hypothesize that this is due to the information bottleneck caused by using a small discrete set of symbols $\Gamma$ in both models, a limitation we hope to address in future work. The interested reader can find experiments for additional model sizes in Appendix E. For all models we find that SG score is highly variable and uncorrelated to perplexity, corroborating findings by Hu et al. (2020). In fact, when we inspected all randomly searched LSTMs, we found that it is sometimes able to attain scores higher than 0.48 (see Appendix E for details). From this we conclude that improving syntax generalization on natural language remains elusive for all stack RNNs we tested, and that we may need to look beyond cross-entropy/perplexity as a training criterion.

## 6   CONCLUSION

The Renormalizing NS-RNN (RNS-RNN) builds upon the strengths of the NS-RNN by letting stack action weights remain unnormalized and providing information about PDA states to the controller. Both of these changes substantially improve learning, allowing the RNS-RNN to surpass other stack RNNs on a range of CFL modeling tasks. Our memory-limited version of the RNS-RNN is a crucial modification towards practical use on natural language. We tested this model on the Penn Treebank, although we did not see performance improvements with the model sizes we were able to test, and in fact no stack RNNs excel in terms of syntactic generalization. We are encouraged by the RNS-RNN's large improvements on CFL tasks and leave improvements on natural language to future work.

## Reproducibility Statement

In order to foster reproducibility, we have released all code and scripts used to generate our experimental results and figures at `https://github.com/bdusell/nondeterministic-stack-rnn`. To ensure that others can replicate our software environment, we developed and ran our code in a Docker container, whose image definition is included in the code repository (see the README for more details). The repository includes the original commands we used to run our experiments, scripts for downloading and preprocessing the PTB dataset of Mikolov et al. (2011), and the test suite definitions needed to compute SG scores. All experimental settings for the CFL experiments are described in §4 and our previous paper (DuSell & Chiang, 2020), and all experimental settings for the PTB experiments may be found in §5.2 and Appendix D.

## Acknowledgements

This research was supported in part by a Google Faculty Research Award to Chiang.

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

# A  BASELINE STACK RNNs

## A.1  STRATIFICATION STACK

We implement the stratification stack of Grefenstette et al. (2015) with the following equations. In the original definition, the controller produces a hidden state $\mathbf{h}_t$ and a separate output $\mathbf{o}'_t$ that is used to compute $a_t$ and $\mathbf{y}_t$, but for simplicity and parity with DuSell & Chiang (2020), we set $\mathbf{o}'_t = \mathbf{h}_t$. Let $m = |\mathbf{v}_t|$ be the stack embedding size.

$$a_t = \text{ACTIONS}(\mathbf{h}_t) = (u_t, d_t, \mathbf{v}_t)$$
$$u_t = \sigma(\mathbf{W}_{\text{hu}}\mathbf{h}_t + \mathbf{b}_{\text{hu}})$$
$$d_t = \sigma(\mathbf{W}_{\text{hd}}\mathbf{h}_t + \mathbf{b}_{\text{hd}})$$
$$\mathbf{v}_t = \tanh(\mathbf{W}_{\text{hv}}\mathbf{h}_t + \mathbf{b}_{\text{hv}})$$
$$s_t = \text{STACK}(s_{t-1}, a_{t-1}) = (V_t, \mathbf{s}_t)$$
$$V_t[i] = \begin{cases} V_{t-1}[i] & 1 \le i < t \\ \mathbf{v}_t & i = t \end{cases}$$
$$\mathbf{s}_t[i] = \begin{cases} \max(0, \mathbf{s}_{t-1}[i] - \max(0, u_t - \sum_{j=i+1}^{t-1} \mathbf{s}_{t-1}[j])) & 1 \le i < t \\ d_t & i = t \end{cases}$$
$$V_0 \text{ is a } 0 \times m \text{ matrix}$$
$$\mathbf{s}_0 \text{ is a vector of size } 0$$
$$\mathbf{r}_t = \text{READING}(s_t) = \sum_{i=1}^{t} (\min(\mathbf{s}_t[i], \max(0, 1 - \sum_{j=i+1}^{t} \mathbf{s}_t[j]))) \cdot V_t[i]$$

Yogatama et al. (2018) noted that the strafication stack can implement multiple pops per time step by allowing $u_t > 1$, although the push action immediately following would still be conditioned on the previous stack top $\mathbf{r}_t$. Hao et al. (2018) augmented this model with differentiable queues that allow it to buffer input and output and act as a transducer. Merrill et al. (2019) experimented with variations of this model where $u_t = 1$ and $d_t \in (0, 4)$, $d_t = 1$ and $u_t = (0, 4)$, and $u_t \in (0, 4)$ and $d_t \in (0, 1)$.

## A.2  SUPERPOSITION STACK

We implement the superposition stack of Joulin & Mikolov (2015) with the following equations. We deviate slightly from the original definition by adding the bias terms $\mathbf{b}_{\text{ha}}$ and $\mathbf{b}_{\text{hv}}$. The original definition also connects the controller to multiple stacks that push *scalars*; instead, we push a vector to a single stack, which is equivalent to multiple scalar stacks whose push/pop actions are synchronized. The original definition includes the top $k$ stack elements in the stack reading, but we only include the

top element. We also treat the value of the bottom of the stack as 0 instead of $-1$.

$$a_t = \text{ACTIONS}(\mathbf{h}_t) = (\mathbf{a}_t, \mathbf{v}_t)$$

$$\mathbf{a}_t = \begin{bmatrix} a_t^{\text{push}} \\ a_t^{\text{noop}} \\ a_t^{\text{pop}} \end{bmatrix} = \text{softmax}(\mathbf{W}_{\text{ha}}\mathbf{h}_t + \mathbf{b}_{\text{ha}})$$

$$\mathbf{v}_t = \sigma(\mathbf{W}_{\text{hv}}\mathbf{h}_t + \mathbf{b}_{\text{hv}})$$

$$\text{STACK}(s_{t-1}, a_{t-1}) = s_t$$

$$s_t[i] = \begin{cases} \mathbf{v}_t & i = 0 \\ a_t^{\text{push}} s_{t-1}[i-1] + a_t^{\text{noop}} s_{t-1}[i] + a_t^{\text{pop}} s_{t-1}[i+1] & 0 < i < t \\ \mathbf{0} & i = t \end{cases}$$

$$\mathbf{r}_t = \text{READING}(s_t) = s_t[1]$$

Yogatama et al. (2018) developed an extension to this model called the Multipop Adaptive Computation Stack that executes a variable number of pops per time step, up to a fixed limit $K$. They also restricted the stack to a maximum size of 10 elements, where the bottom element of a full stack is discarded when a new element is pushed; in other words, $s_t[i] = \mathbf{0}$ for $i > K$. Suzgun et al. (2019) experimented with a modification of the parameterization of $\mathbf{a}_t$ and different softmax operators for normalizing the weights used to compute $\mathbf{a}_t$.

## B  DETAILS OF FORMAL LANGUAGE EXPERIMENTS

For every training run, we sample a training set of 10,000 strings from the PCFG, with lengths drawn uniformly from $[40, 80]$. Similarly, we sample a validation set of 1,000 strings with lengths drawn uniformly from $[40, 80]$. For each task, we sample a test set of 100 strings per length for each length in $[40, 100]$. Whereas the training and validation sets are randomized for each experiment, the test sets are the same across all models and random restarts.

In all cases, the LSTM has a single layer with 20 hidden units. We grid-search the initial learning rate from $\{0.01, 0.005, 0.001, 0.0005\}$. For Gref and JM, we search for stack vector element sizes in $\{2, 20, 40\}$ (the pushed vector in JM is learned). For the NS models, we manually choose a small number of PDA states and stack symbol types based on how we would expect a PDA to solve the task. For marked reversal, unmarked reversal, and Dyck, we use $|Q| = 2$ and $|\Gamma| = 3$; and for padded reversal and hardest CFL, we use $|Q| = 3$ and $|\Gamma| = 3$. For each hyperparameter setting searched, we run five random restarts. For each type of model, we select the model with the lowest difference in cross-entropy between the model and true distribution on the validation set. We use the same initialization and optimization settings as in our earlier paper (DuSell & Chiang, 2020) and train for a maximum of 200 epochs.

## C  WALL-CLOCK TRAINING TIME

We report wall-clock execution time for each model on the marked reversal task in Table 2. We ran the LSTM, Gref, and JM models in CPU mode, as this was faster than running on GPU due to the small model size. We ran experiments for the NS models in GPU mode on a pool of the following NVIDIA GPU models, automatically selected based on availability: GeForce GTX TITAN X, TITAN X (Pascal), and GeForce GTX 1080 Ti.

## D  DETAILS OF NATURAL LANGUAGE EXPERIMENTS

The hyperparameters for our baseline LSTM, initialization, and optimization scheme are based on the unregularized LSTM experiments of Semeniuta et al. (2016). We train all models using simple stochastic gradient descent (SGD) as recommended by prior language modeling work (Merity et al., 2018) and truncated BPTT with a sequence length of 35. For all models, we use a minibatch size of 32. We randomly initialize all parameters uniformly from the interval $[-0.05, 0.05]$. We divide the

Table 2: Wall-clock execution time for each model on the marked reversal task, measured in seconds per epoch of training (averaged over all epochs). The speed of the NS models is roughly the same; there is some variation here due to differences in training data and GPU model.

| Model | Time per epoch (s) |
|---|---|
| LSTM | 51 |
| Gref | 801 |
| JM | 169 |
| NS | 1022 |
| NS+S | 980 |
| NS+U | 960 |
| NS+S+U | 1060 |

learning rate by 1.5 whenever the validation perplexity does not improve, and we stop training after 2 epochs of no improvement in validation perplexity.

For each model, we randomly search for initial learning rate and gradient clipping threshold; we report results for the model with the best validation perplexity out of 10 randomly searched models. The learning rate, which is divided by batch size and sequence length, is drawn from a log-uniform distribution over $[1, 100]$, and the gradient clipping threshold, which is multiplied by batch size and sequence length, is drawn from a log-uniform distribution over $[1 \times 10^{-5}, 1 \times 10^{-3}]$. (We scale the learning rate and gradient clipping threshold this way because, under our implementation, sequence length and batch size can vary when the data set is not evenly divisible by the prescribed values. Other language modeling papers follow a different scaling convention for these two hyperparameters, typically scaling the learning rate by sequence length but not by batch size, and not rescaling the gradient clipping threshold. Under this convention the learning rate would be drawn from $[0.03125, 3.125]$ and the gradient clipping threshold from $[0.0112, 1.12]$.)

## E    ADDITIONAL RESULTS FOR NATURAL LANGUAGE EXPERIMENTS

In Table 3 we show additional experimental results on the Penn Treebank. In Table 4 we show the same experiments with SG score broken down by syntactic "circuit" as defined by Hu et al. (2020), offering a more fine-grained look at the classes of errors the models make. We see that SG score is highly variable and does not follow the same trends as perplexity. In Figure 4, we plot SG score vs. test perplexity for all 10 random restarts of an LSTM and an RNS-RNN. We see that many of the models that were not selected actually have a much higher SG score (even above 0.48), suggesting that the standard validation perplexity criterion is a poor choice for syntactic generalization.

Table 3: Language modeling results on PTB, measured by perplexity and SG score, with additional experiments included.

| Model | # Params | Val | Test | SG Score |
|---|---|---|---|---|
| LSTM, 256 units | 5,656,336 | 125.78 | 120.95 | 0.433 |
| LSTM, 258 units | 5,704,576 | 122.08 | 118.20 | 0.420 |
| LSTM, 267 units | 5,922,448 | 125.20 | 120.22 | 0.437 |
| JM (push hidden state), 247 units | 5,684,828 | **121.24** | **115.35** | 0.387 |
| JM (push learned), $|\mathbf{v}_t| = 22$ | 5,685,289 | 122.87 | 117.93 | 0.431 |
| NS, $|Q| = 1$, $|\Gamma| = 2$ | 5,660,954 | 126.10 | 122.62 | 0.414 |
| NS, $|Q| = 1$, $|\Gamma| = 3$ | 5,664,805 | 123.41 | 119.25 | 0.430 |
| NS, $|Q| = 1$, $|\Gamma| = 4$ | 5,669,684 | 121.66 | 117.91 | 0.432 |
| NS, $|Q| = 1$, $|\Gamma| = 5$ | 5,675,591 | 123.01 | 119.54 | 0.452 |
| NS, $|Q| = 1$, $|\Gamma| = 6$ | 5,682,526 | 129.94 | 125.45 | 0.432 |
| NS, $|Q| = 1$, $|\Gamma| = 7$ | 5,690,489 | 126.11 | 121.94 | 0.443 |
| NS, $|Q| = 1$, $|\Gamma| = 11$ | 5,732,621 | 129.11 | 124.98 | 0.431 |
| NS, $|Q| = 2$, $|\Gamma| = 2$ | 5,668,664 | 128.16 | 123.52 | 0.412 |
| NS, $|Q| = 2$, $|\Gamma| = 3$ | 5,680,996 | 129.51 | 126.00 | **0.471** |
| NS, $|Q| = 2$, $|\Gamma| = 4$ | 5,697,440 | 124.28 | 120.18 | 0.433 |
| NS, $|Q| = 2$, $|\Gamma| = 5$ | 5,717,996 | 124.24 | 119.34 | 0.429 |
| NS, $|Q| = 3$, $|\Gamma| = 2$ | 5,681,514 | 125.32 | 120.62 | 0.470 |
| NS, $|Q| = 3$, $|\Gamma| = 3$ | 5,707,981 | 122.96 | 118.89 | 0.420 |
| NS, $|Q| = 3$, $|\Gamma| = 4$ | 5,743,700 | 126.71 | 122.53 | 0.447 |
| RNS, $|Q| = 1$, $|\Gamma| = 2$ | 5,660,954 | 122.64 | 117.56 | 0.435 |
| RNS, $|Q| = 1$, $|\Gamma| = 3$ | 5,664,805 | 121.83 | 116.46 | 0.430 |
| RNS, $|Q| = 1$, $|\Gamma| = 4$ | 5,669,684 | 127.99 | 123.06 | 0.437 |
| RNS, $|Q| = 1$, $|\Gamma| = 5$ | 5,675,591 | 126.41 | 122.25 | 0.441 |
| RNS, $|Q| = 1$, $|\Gamma| = 6$ | 5,682,526 | 122.57 | 117.79 | 0.416 |
| RNS, $|Q| = 1$, $|\Gamma| = 7$ | 5,690,489 | 123.51 | 120.48 | 0.430 |
| RNS, $|Q| = 1$, $|\Gamma| = 11$ | 5,732,621 | 127.21 | 121.84 | 0.386 |
| RNS, $|Q| = 2$, $|\Gamma| = 2$ | 5,670,712 | 122.11 | 117.22 | 0.399 |
| RNS, $|Q| = 2$, $|\Gamma| = 3$ | 5,684,068 | 131.46 | 127.57 | 0.463 |
| RNS, $|Q| = 2$, $|\Gamma| = 4$ | 5,701,536 | 124.96 | 121.61 | 0.431 |
| RNS, $|Q| = 2$, $|\Gamma| = 5$ | 5,723,116 | 122.92 | 117.87 | 0.423 |
| RNS, $|Q| = 3$, $|\Gamma| = 2$ | 5,685,610 | 129.48 | 124.66 | 0.433 |
| RNS, $|Q| = 3$, $|\Gamma| = 3$ | 5,714,125 | 127.57 | 123.00 | 0.434 |
| RNS, $|Q| = 3$, $|\Gamma| = 4$ | 5,751,892 | 122.67 | 118.09 | 0.408 |

Table 4: SG scores broken down by circuit. Agr. = Agreement, Lic. = Licensing, GPE = Garden-Path Effects, GSE = Gross Syntactic Expectation, CE = Center Embedding, LDD = Long-Distance Dependencies.

| Model | Agr. | Lic. | GPE | GSE | CE | LDD |
|---|---|---|---|---|---|---|
| LSTM, 256 units | 0.667 | 0.446 | 0.330 | 0.397 | 0.482 | 0.414 |
| LSTM, 258 units | 0.658 | 0.447 | 0.335 | 0.375 | 0.518 | 0.357 |
| LSTM, 267 units | 0.667 | 0.497 | 0.343 | 0.446 | 0.411 | 0.350 |
| JM (push hidden state) | 0.640 | 0.408 | 0.296 | 0.310 | 0.464 | 0.352 |
| JM (push learned) | 0.684 | 0.439 | 0.340 | 0.408 | 0.482 | 0.395 |
| NS, $|Q| = 1, |\Gamma| = 2$ | 0.588 | 0.452 | 0.298 | 0.391 | 0.339 | 0.418 |
| NS, $|Q| = 1, |\Gamma| = 3$ | 0.623 | 0.467 | 0.400 | 0.413 | 0.393 | 0.354 |
| NS, $|Q| = 1, |\Gamma| = 4$ | 0.640 | 0.497 | 0.331 | 0.375 | 0.571 | 0.340 |
| NS, $|Q| = 1, |\Gamma| = 5$ | 0.605 | 0.514 | 0.394 | 0.413 | 0.589 | 0.344 |
| NS, $|Q| = 1, |\Gamma| = 6$ | 0.632 | 0.424 | 0.408 | 0.391 | 0.464 | 0.399 |
| NS, $|Q| = 1, |\Gamma| = 7$ | 0.719 | 0.470 | 0.351 | 0.473 | 0.500 | 0.344 |
| NS, $|Q| = 1, |\Gamma| = 11$ | 0.640 | 0.432 | 0.329 | 0.424 | 0.500 | 0.413 |
| NS, $|Q| = 2, |\Gamma| = 2$ | 0.702 | 0.388 | 0.329 | 0.446 | 0.446 | 0.371 |
| NS, $|Q| = 2, |\Gamma| = 3$ | 0.658 | 0.527 | 0.367 | 0.446 | 0.518 | 0.411 |
| NS, $|Q| = 2, |\Gamma| = 4$ | 0.632 | 0.464 | 0.345 | 0.386 | 0.518 | 0.387 |
| NS, $|Q| = 2, |\Gamma| = 5$ | 0.711 | 0.464 | 0.307 | 0.413 | 0.518 | 0.355 |
| NS, $|Q| = 3, |\Gamma| = 2$ | 0.711 | 0.528 | 0.349 | 0.435 | 0.518 | 0.406 |
| NS, $|Q| = 3, |\Gamma| = 3$ | 0.746 | 0.439 | 0.316 | 0.375 | 0.411 | 0.376 |
| NS, $|Q| = 3, |\Gamma| = 4$ | 0.702 | 0.450 | 0.364 | 0.484 | 0.536 | 0.369 |
| RNS, $|Q| = 1, |\Gamma| = 2$ | 0.702 | 0.460 | 0.280 | 0.451 | 0.464 | 0.404 |
| RNS, $|Q| = 1, |\Gamma| = 3$ | 0.649 | 0.427 | 0.438 | 0.418 | 0.446 | 0.347 |
| RNS, $|Q| = 1, |\Gamma| = 4$ | 0.658 | 0.412 | 0.342 | 0.565 | 0.339 | 0.418 |
| RNS, $|Q| = 1, |\Gamma| = 5$ | 0.728 | 0.449 | 0.370 | 0.429 | 0.482 | 0.371 |
| RNS, $|Q| = 1, |\Gamma| = 6$ | 0.614 | 0.422 | 0.314 | 0.435 | 0.518 | 0.377 |
| RNS, $|Q| = 1, |\Gamma| = 7$ | 0.649 | 0.460 | 0.374 | 0.337 | 0.411 | 0.404 |
| RNS, $|Q| = 1, |\Gamma| = 11$ | 0.614 | 0.447 | 0.291 | 0.266 | 0.446 | 0.338 |
| RNS, $|Q| = 2, |\Gamma| = 2$ | 0.649 | 0.417 | 0.365 | 0.375 | 0.339 | 0.334 |
| RNS, $|Q| = 2, |\Gamma| = 3$ | 0.640 | 0.474 | 0.411 | 0.446 | 0.554 | 0.408 |
| RNS, $|Q| = 2, |\Gamma| = 4$ | 0.658 | 0.469 | 0.336 | 0.326 | 0.500 | 0.403 |
| RNS, $|Q| = 2, |\Gamma| = 5$ | 0.693 | 0.420 | 0.339 | 0.370 | 0.607 | 0.376 |
| RNS, $|Q| = 3, |\Gamma| = 2$ | 0.579 | 0.435 | 0.295 | 0.440 | 0.554 | 0.445 |
| RNS, $|Q| = 3, |\Gamma| = 3$ | 0.632 | 0.444 | 0.356 | 0.418 | 0.482 | 0.403 |
| RNS, $|Q| = 3, |\Gamma| = 4$ | 0.588 | 0.427 | 0.342 | 0.353 | 0.482 | 0.373 |

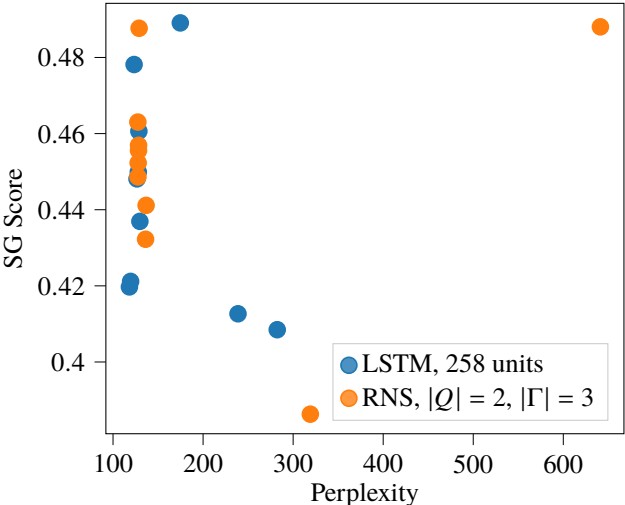

Figure 4: SG score vs. test perplexity, shown on all 10 random restarts for an LSTM and an RNS-RNN. SG score is uncorrelated with perplexity, and models that narrowly miss out on having the best perplexity often have much higher SG scores.

