# OpenReview forum: "Learning Hierarchical Structures with Differentiable Nondeterministic Stacks"
_ICLR.cc/2022/Conference — ICLR 2022 Spotlight_

### Official Review · Reviewer_Cbcr · 2021-11-02

**Correctness:** 3
**Technical Novelty And Significance:** 4
**Empirical Novelty And Significance:** 4
**Recommendation:** 8
**Confidence:** 2

**Main Review:**

# Strength

- It is a very interesting work on differentiable non-deterministic PDA in terms of the theoretical view point. Although the gains are not large and computationally demanding, the proposed enhancements over prior work are very interesting in that they allow its application to the real World data, i.e., PTB.

# Weakness

- The gains are not large, but meaningful improvements are observed in the synthetic data set and the gains in SG scores would be good enough, I believe.

- I feel the bound for stack could be treated as one of the contribution of this work. I'd suggest moving the description of section 5.1 into section 3 to emphasize the nice contribution.

- I'd like to see further analysis to indicate whether the proposed model could learn hierarchical structures as indicated in the title under PTB. Probably manual assessments would be good enough to show some evidence.

**Summary Of The Paper:**

This work presents various improvements over the differentiable non-deterministic push-down automaton. Basic idea of non-deterministic push-down automaton is to explore all the space for state id with stack with push/replace/pop operators with dynamic programming. Given the prior work of DuSell & Chiang (2020), this work has three contributions:

- Do not normalize for transition, which is similar to migrating from maximum entropy Markov model to conditional random filed, so that the model is not biased by local transitions, but globally normalilzed.

- Consider the joint distribution of stack symbol and state, not only stack symbol, so that the controller can differentiate by the current state of the PDA.

- The model is not space efficient in that the stack is unbound, and thus, consumes large memory. This work proposes to limit the stack memory so that the model can run on the real World data, e.g., PTB.

Experimental results on synthetic data show large improvement in terms of per-symbol cross-entropy. The perplexities on PTB do not achieve better results when compared with other simpler variants or LSTM. However, the scores on Syntactic Generalization show some gains.

**Summary Of The Review:**

The proposed method is sound and the experiments are designed nicely to prove the improvements of this work. Although the gains are not significant, I'd rather like to see this paper accepted given that it is an interesting contribution to computational linguistics, and might have an impact in the future.

---

> ### Author Response · Authors · 2021-11-17
> **On incremental execution and analysis of learned structures**
>
> Dear Reviewer Cbcr,
>
> Thank you very much for your feedback. We agree that the incremental execution technique is an important contribution, and we will consider reorganizing its presentation, at least by emphasizing it more in the introduction. As for analyzing hierarchical structures learned on PTB, we did perform some analysis by running Viterbi decoding on the stack WFA in order to determine the most likely sequence of PDA transitions leading up to a given time step. We did not have space to include this in the current revision of the paper, but we will include some examples if we have space. Qualitatively, although in some cases the learned operations looked like a plausible parse of the sentence, we found that the PDA often did not empty the stack by the time it reached the end of the sentence. We think this is because the PDA requires more information bandwidth in the stack to carry a useful signal back in time (as mentioned in Section 5.2, we would like to address this in future work), but this will require more analysis.

---

> ### Author Response · Authors · 2021-11-21
> **Paper revisions**
>
> Dear Reviewer Cbcr,
>
> Thank you again for your feedback. We have revised our paper to emphasize the significance of the memory-restricted RNS-RNN at the end of Section 1.

---

### Official Review · Reviewer_THSf · 2021-11-02

**Correctness:** 3
**Technical Novelty And Significance:** 2
**Empirical Novelty And Significance:** 2
**Recommendation:** 6
**Confidence:** 3

**Main Review:**

The proposed method solves two defects of NS-RNN. The ideas are technically sound and empirically proved.
However, I have several concerns:
1. The RNS-RNN is based on a weighted PDA, which has a finite set of states. This setting limits the capacity and expressiveness of the proposed model, especially when modern deep learning methods usually use very large hidden states to achieve high expressiveness.
2. The RNS-RNN achieves a marginally better result on PTB language modeling task while comparing to NS-RNN, but still falling far behind any strong baselines.
3. The formal language tasks are toyish. It would be interesting to test the proposed method on less toyish formal language tasks, for example, logical inference (Bowman et al., 2015) and ListOps (Nangia and Bowman, 2018).

Missing reference:
- Shen, Yikang, et al. "Ordered Memory." Advances in Neural Information Processing Systems 32 (2019): 5037-5048.

**Summary Of The Paper:**

The paper proposes a new stack-augmented RNN, RNS-RNN that includes two modifications to the Nondeterministic Stack RNN:
1. RNS-RNN uses unnormalized transition weights to avoid probability vanishing problems, such that gradient can be easier backpropagated to certain decisions at a previous time step.
2. RNS-RNN feeds the hidden state y into the controller while reading the stack, this allows the controller has more complete information to make decisions.

**Summary Of The Review:**

Overall the paper proposed useful improvements to the NS-RNN. However, the novelty and impact of this paper are not significant enough.

---

> ### Author Response · Authors · 2021-11-17
> **On model capacity and choices of tasks**
>
> Dear Reviewer THSf,
>
> Thank you very much for your comments and pointers to related work.
>
> Response to comment 1: Fortunately, like the stratification and superposition stack RNNs, the (R)NS-RNN connects a stack module to an LSTM and does not replace any elements of the LSTM. The presence of the PDA data structure (what we call the "stack WFA") gives the LSTM a hierarchical inductive bias, but it does not subtract from the LSTM's capacity or expressive power. The hidden state $(\mathbf{h}_t, \mathbf{c}_t)$ and set of PDA states $Q$ are separate things, and the size of one does not limit the other.
>
> Response to comment 2: Please see our response to comment 2 in our response to Reviewer vA7w.
>
> Response to comment 3: Thank you for the pointers to these two tasks; we agree that they would be appropriate benchmarks for the RNS-RNN. We did consider running experiments on ListOps but decided not to, for three reasons. First, the task was effectively solved in Havrylov et al. (2019). Second, solving it would require at least $|\Gamma| = 10$, which is rather computationally expensive (we hope to alleviate this problem in future work and may evaluate on ListOps then).
> Third, it seems that ListOps (and Bowman 2015) can be solved with a deterministic stack model, while our focus here is on tasks that require nondeterminism.
> We believed it was important to test the RNS-RNN on the same tasks as the original NS-RNN paper (DuSell \& Chiang, 2020) in order to provide a direct comparison. We also believe the unmarked reversal, padded reversal, and especially the hardest CFL tasks provide crucial evidence that the RNS-RNN improves on tasks with ambiguous syntax, which is the phenomenon we were targeting.
>
> Thank you for pointing us to the paper "Ordered Memory." We will be sure to mention it as related work on stack-structured memory in our paper.

---

> ### Author Response · Authors · 2021-11-21
> **Paper revisions**
>
> Dear Reviewer THSf,
>
> Thank you again for your feedback. We have revised our paper in response to your comments. Specifically:
>
> * We have updated Section 2.4 to make it clearer that the stack WFA is a separate module from the LSTM controller.
> * We have clarified our conclusions from the natural language experiments in Section 5.2.
> * We have included references to "Ordered Memory" and related work on shift-reduce parsing in Section 1.

---

> > ### Comment · Reviewer_THSf · 2021-12-01
> > **Thanks**
> >
> > Thanks for your clarification. The author addressed my question on capacity. Considering the opinion of other reviewers, I acknowledge that the contributions of this paper are interesting for researchers in this field. I have adjusted my recommendation score to 6.

---

### Official Review · Reviewer_vA7w · 2021-11-04

**Correctness:** 4
**Technical Novelty And Significance:** 3
**Empirical Novelty And Significance:** 2
**Recommendation:** 8
**Confidence:** 3

**Main Review:**

=quick pros=
- proposed changes are simple, clear, and appear to improve the model on some synthetic tasks (even significantly for one of them).

=quick cons=
- the model does not seem very good on natural language tasks. However, if my understanding of stack-rnn research is correct, this is to be expected.

=full review=

Note: I have reviewed this paper before, and appreciate that the authors have added experiments and metrics as requested.

I am adding some presentation comments, some of which are new, and some of which I think are important (even if they are minor), please take a look through them! Note: I recognise you responded to some of these questions in the previous review round, but I would like them answered in the paper too! :)

==some comments==

1. I find the first change ((1) in the summary) interesting, if simple, and the second ((2) in the summary) a little odd: intuitively, because the RNN outputs a next-move distribution for *every* current top-token and current-state pair, why would it need to know what the current one *actually* is? I am glad that you evaluated each change independently as well as together, and so I can be convinced that the change is necessary (e.g. through the results on unmarked reversal), but still I do not fully understand it.  I would appreciate a greater discussion of *why* this change is good here - why does it help despite my intuition? (The explanation of section 3.2 is not working for me: following my intuition here, I still don't see the importance of the state-read.)

2. It is not clear what the SG score adds to the discussion, or what I should conclude from it, if at all. It would help to add a definition of the SG score to the paper (as done for cross entropy), or at least some greater discussion if this is not possible, and provide some intuition on what it means.

3 (important!). I appreciate that the authors have been very explicit in all of their constructions and all details of their experiments, I find this very valuable. However, I would also like to note that as someone not familiar with the NS-RNN, I do struggle with the formulas. In particular in equation 2 I would like some more explanation on the [i->t] inputs - it seems there are jumps over several time steps?? This is a very confusing point that could do with some plain-english explanation.

(3b. Maybe personal preference: Another thing to note in this direction is that the last paragraph of the introduction describes the RNS-RNN according to how it differs from the NS-RNN, but for anyone who is not already familiar with the NS-RNN, it is not useful. If possible, I think adding another paragraph that more deeply explains---in non-technical terms (i.e., no formulas)---what exactly an NS-RNN is doing would be very helpful. (Approximately, though will need rephrasing: defining and maintaining a distribution over all possible configurations of a non-deterministic PDA, by computing at each step: for each state and stack-top combination, a distribution over all possible next-state and stack-actions. And from this, each configuration's probability is the sum of the products of each sequence of transitions and state-actions that get to that configuration). )

4. In figure 3, it would be nice to also see this plot for NS, JM, Gref, and NS+U! Maybe in the appendix at least?

5. Section 4, paragraph after Hardest CFL: "...hardest CFL requiring the most." I don't understand what this means - what does it mean for one non-deterministic language to require "more" non-determinism than another?

==some questions==
1. In unmarked reversal, it is interesting to note that each of the changes alone seems to barely improve on the NS-RNN, but that together they improve the model significantly. Could you add a discussion, sharing any intuition on why this is? (Section 3.2 is not enough - at that point I still do not feel I understand why the second change is needed, as noted above in my review).

2. In figure 3, there is a white line in the middle of the image for both models, suggesting that it is not learning the correct action at the point where the reversing 'begins'. Could you add some discussion explaining this?

3. Can you explain the intuition behind adding the EOS tokens to the end of each sample (as opposed to the experiments in the original NS-RNN paper)? I understand that it makes the model define a proper distribution (as opposed to weights) over the sequences, but I don't understand why this is important for these evaluations. (Is predicting the EOS hard for any of these tasks, or something like that? I see that you comment that it makes things harder for the baseline NS, do you have ideas on why?)


==typos, presentation, minor comments==
1. Section 2.3 (superposition) is hard to understand, an analogy (similar to cake for stratification) could help. If one is not available, at least a concrete example, e.g., what happens if I have a stack of depth 5 and push/pop/noop with probabilities 0.3/0.5/0.2 ?
2. Section 2.4, second paragraph, the description of the possible operations: the paragraph says that it is showing Op(Gamma), but it seems you are showing Q \times Op(Gamma). Consider removing the "r"'s from each operation, or introducing the list as Q\times Op(Gamma).
3. Formal languages results: Can you explain the higher difference in cross entropy, for all models and languages, on the shorter (but in train range) sequences?


**Summary Of The Paper:**

This work continues a recent work on nondeterministic stack RNNs (in which an RNN controls a nondeterministic pushdown automaton, which I will refer to here as an N-PDA), proposing two changes to the architecture which create an RNS-RNN (renormalised NS-RNN) as follows:
1. Where in the NS-RNN the RNN-controller outputs a set of distributions over the next possible PDA actions (one distribution for every current state and stack-top pair), the RNS-RNN outputs scores over these actions. This is achieved simply by not softmaxing the scores that are anyway computed in the NS-RNN. The goal/effect is that the score of each (current state,current stack) pair being maintained in the N-PDA does not tend to zero as the input sequence length increases (as it would when probabilities are used), and in fact can even be increased at certain transitions.
2. Where in the NS-RNN the RNN-controller receives as part of its input at each time step a distribution over the current stack top token, it now also receives a distribution over the current PDA state. (These are normalised back to distributions here, normalised from the scores kept in the N-PDA.)

The authors evaluate the RNS-RNN against the NS-RNN and several other architectures, showing its success on various formal languages and even a greater success on a "hardest" context free grammar.

The NS-RNN requires quadratic space and cubic time in the input sequence length*, which is impractical for natural language processing. The authors propose a memory-limited version which reduces the complexity to linear time and space in the input sequence length, and discuss truncated BPTT for this version. They evaluate a memory limited RNS-RNN on the Penn Treebank.

*Because it is maintaining the probabilities of all possible configurations of non-deterministic PDA on the current input---in fact we are lucky this is not exponential! A dynamic algorithm by Lang is used to maintain the PDA, I am not sure of the details


**Summary Of The Review:**

This paper makes straightforward (this is good) changes to an existing stack-augmented RNN model, the NS-RNN, evaluating the new model on some formal languages and even one natural language dataset. The improvements are modest (and on the natural language dataset, unclear), but sufficient to be of interest. The paper is well written, though it could do with some expansions and clarifications as elaborated in the main review.

---

> ### Author Response · Authors · 2021-11-17
> **Responses to comments**
>
> Dear Reviewer vA7w,
>
> Once again, thank you very much for your thoughtful feedback and suggestions. We welcome your suggestions on clarity, many of which we were unable to address in this revision simply due to space constraints.
>
> Response to comment 1: The reason including the PDA state in the stack reading is important can be summarized by the motivating example on unmarked reversal given in Section 3.2, and indeed, this is the task for which adding PDA states makes the most difference. When reading the first half of a string in the unmarked reversal language, the RNS-RNN only needs to predict a uniform distribution, but when reading the second half, its prediction depends on the current top stack symbol. However, the model does not know where the (unmarked) split point is, so it relies on a PDA with at least two states to guess where it is nondeterministically. So, the solution the model should learn is to interpolate between the uniform and top-informed distributions, weighted by the probabilities of being in state 1 and state 2. The RNN controller by itself cannot simulate the PDA state effectively, because the state is entangled with the distribution over stacks in the stack WFA. So, knowledge of the current PDA state is required to make optimal predictions. As you point out, since the RNN controller outputs a conditional distribution for each state/stack symbol pair, passing the state back to the controller is not necessary for generating the optimal stack operations at the next timestep, but it *is* necessary for predicting the optimal $y_t$, which depends on $\mathbf{h}_t$. We will elaborate on this in Section 3.2.
>
> Response to comment 2: The SG score consists of a fixed set of targeted syntax tasks in English inspired by psycholinguistics research. For example, it tests that the model assigns higher probability to "The farmer near the clerks knows many people" than "The farmer near the clerks know many people". SG score is an accuracy score averaged across multiple types of syntactic test. As noted by Hu et al. (2020), perplexity does not correlate with SG score, which suggests that if a model has low perplexity, this does not necessarily tell the whole story of how well it generalizes on syntactic structures. We believe this kind of evaluation is particularly important for stack RNNs. In Tables 1 and 3, we reproduce an earlier finding from Yogatama et al. (2018) that shows that JM can get lower perplexity than an LSTM, but this does not translate to a better SG score, which varies a lot across the board. Our takeaway from this is that improving syntax generalization on natural language remains elusive for all stack RNNs we tested, and that we may need to look beyond cross-entropy/perplexity as a training criterion for learning this behavior. We will clarify this in our paper.
>
> Response to comment 3: We will gladly use the extra space in our revision to provide a more intuitive, self-contained introduction to Lang's algorithm and the original NS-RNN model. It may help to note that the weight $\gamma[i \rightarrow t][q, x \rightarrow r, y]$ has a particular meaning. Let the triple $(i, q, x)$ represent a PDA configuration where the PDA is in state $q$ with $x$ on top of the stack at timestep $i$. The value $\gamma[i \rightarrow t][q, x \rightarrow r, y]$ represents the weight of the PDA going from configuration $(i, q, x)$ to $(t, r, y)$, where the only difference in the stack is that a single $y$ symbol has been pushed onto the $x$, and the $x$ has never been modified. There are many sequences of operations that can connect these configurations -- and yes, they can span multiple timesteps. For example, (push $z$, pop) and (push $z$, repl $x$, pop) are sequences that contribute to this weight, but (pop, push $x$) is not because it tampers with $x$.
>
> As you pointed out, Lang's algorithm helps us marginalize over an exponential number of PDA runs (run = sequence of operations) with only polynomial time and space complexity. It is a dynamic programming algorithm that takes advantage of a couple of structural similarities between runs. First, multiple runs can result in the same PDA state and stack. Second, a stack of height $k$ must have been derived from a stack of height $k-1$, so we can encode the stack of height $k$ using only its top symbol and pointers to several stacks of height $k-1$. What we end up with is a directed graph of polynomial size, where each path, of which there are exponentially many, encodes a stack. In the paper we describe this graph as a finite automaton ("stack WFA") encoding the language of possible stacks, which is regular. The chapter "Context-free Languages and Pushdown Automata" (Autebert et al., 1997) provides a proof in Section 5.3 that the language of stacks is regular, and Lang's algorithm gives an explicit construction for a finite automaton that recognizes this language of stacks.
>
> (to be continued)

---

> > ### Author Response · Authors · 2021-11-17
> > **Responses to comments and questions**
> >
> > (continued)
> >
> > Response to comment 4: Yes, this would be possible and is a good suggestion, although it would require additional code and experiments. We might include this if we have time.
> >
> > Response to comment 5: See Appendix A of DuSell & Chiang (2020), and our response to Reviewer cb1a.
> >
> > Response to question 1: This is a good question. We believe that the unnormalized weights allow the model to propagate signals more easily across time as in marked reversal, and the states help for the reasons given in our response to comment 1. It is indeed interesting that both modifications at the same time are required. Without unnormalized weights, the model does not find a good solution during training; without PDA states, the model does not have enough information to make optimal decisions. We will add this discussion to the paper.
> >
> > Response to question 2: For the middle timestep, we considered "replace" to be the "correct" action, since that is what a hand-coded PDA would do. Apparently the model learns to execute a different action at this timestep, without affecting the results. We will make a note of this in the paper.
> >
> > Response to question 3: We added EOS mainly because we felt that results are easier to interpret when the model defines a probability distribution over strings that sums to 1. This is particularly true for the test sets, because using EOS guarantees that the difference in cross entropy has a lower bound of 0, whereas when EOS is not used, the difference can be negative. DuSell \& Chiang (2020) include a note about this in their Section 5.3. A likely reason that predicting EOS is so difficult for the NS-RNN on unmarked reversal is that it needs to learn a correlation between the two most distant timesteps in the string, something the RNS-RNN is designed to do better.
> >
> > Response to presentation comment 1: Another way to think about the superposition stack is that at each timestep, it computes three new, separate stacks: one with all cells shifted down (push), kept the same (no-op), and shifted up (pop). The three stacks are interpolated element-wise to form the next stack. We will include this remark in the paper if we have space.
> >
> > Response to presentation comment 2: Thank you -- we will address this.
> >
> > Response to presentation comment 3: All models seem to have a preference for strings that lie close to the mean string length. We don't have a specific explanation for this, except that it may simply be "averaging" over the distribution of string lengths in the training data, which is uniform over $[40, 80]$.

---

> ### Author Response · Authors · 2021-11-21
> **Paper revisions**
>
> Dear Reviewer vA7w,
>
> Thank you again for your thorough feedback. We have revised our paper to take your comments and questions into account. Specifically:
>
> * We have adjusted Section 3.2 to more clearly explain the importance of PDA states.
> * We have clarified our conclusions in Section 5.2 in response to comment 2.
> * We have included a more thorough explanation of Lang's algorithm and the NS-RNN, including more intuition of how it represents an exponential number of stacks with only polynomial space. We have also resolved the ambiguity concerning $\mathrm{Op}(\Gamma)$ and $r$.
> * In Section 4, we now refer the reader to Appendix A of DuSell & Chiang (2020) for more details about the Hardest CFL.
> * We have included more discussion of our results on unmarked reversal in Section 4, per our response to question 1.
> * In the caption of Figure 3, we explain the presence of the white band in the middle.
> * We have added a short comment on why EOS makes unmarked reversal harder for the NS-RNN in Section 4.
> * We have updated our explanation of the superposition stack in Section 2.3.

---

> > ### Comment · Reviewer_vA7w · 2021-11-30
> > **Thank you**
> >
> > Thank you for your detailed responses, and I appreciate also seeing updates in the paper. My score remains as it was (positive). All the best!

---

### Official Review · Reviewer_cb1a · 2021-11-08

**Correctness:** 3
**Technical Novelty And Significance:** 4
**Empirical Novelty And Significance:** 4
**Recommendation:** 6
**Confidence:** 3

**Main Review:**

Strengths
- The discussion on previous stack RNNs was a strong segment of the paper as the authors connect the various approaches using common notation.
- The modifications to the NS-RNN are well motivated by appealing to toy problems or an explanation of what happens numerically during training.
- Transitioning the discussion from formal languages to natural languages was a good motivation for introducing the memory-limited technique. Good use of SG score metric instead of just perplexity.

Weaknesses
- It would have been good to explain what Hardest CFL (Greibach, 1973) is in the paper. **[addressed by author response and paper update]**
- I might be missing something, but what is the intuition behind the choice of $Q$ or $\Gamma$ for the models in Table 1? **[addressed by author response and paper update]**

**Summary Of The Paper:**

This paper aims to improve the performance of the Nondeterministic Stack RNN (NS-RNN) using unnormalized positive weights instead of probabilities for stack actions and allowing the model to directly observe the state.

The paper also uses the new NS-RNN for a language modelling task on the Penn Treebank by introducing a memory-limiting technique.

**Summary Of The Review:**

Overall, I think this work is marginally above the acceptance threshold. The improvements made to NS-RNN are well-motivated. While this may seem to be incremental, I think the contribution of the memory-limiting approach enhances the paper's relevance by allowing for language modeling over natural language.

---

> ### Author Response · Authors · 2021-11-17
> **On the Hardest CFL and hyperparameter choices**
>
> Dear Reviewer cb1a,
>
> Thank you very much for your review and constructive comments.
>
> In response to your comment about the Hardest CFL, it may be helpful to look at Appendix A of the NS-RNN paper (DuSell \& Chiang 2020), which provides a brief, intuitive description of the language. We will be sure to add remarks about the theory behind the Hardest CFL in the appendix.
>
> As for our choices for $|Q|$ and $|\Gamma|$ in Table 1, the setting $|Q| = 1$ and $|\Gamma| = 2$ represents minimal capacity in the (R)NS-RNN models and is meant to serve as a baseline for the other settings. The other two settings are meant to test the upper limits of model capacity before computational cost becomes too great. The setting $|Q| = 1$ and $|\Gamma| = 11$ represents the greatest number of stack symbol types we can afford to use, using only one PDA state. We selected the setting $|Q| = 3$ and $|\Gamma| = 4$ by increasing the number of PDA states, and then the number of stack symbol types, until computational cost became too great (recall that the time complexity is $O({|Q|}^4 {|\Gamma|}^3)$, so adding states is more expensive than adding stack symbol types).

---

> > ### Comment · Reviewer_cb1a · 2021-11-30
> > **Thank you**
> >
> > Thank you for the helpful explanation on the choice of hyperparameters and including it in the caption of Table 1.
> >
> > My score remains the same, but I’m excited for this work to be shared with a larger audience!

---

> ### Author Response · Authors · 2021-11-20
> **Paper revisions**
>
> Dear Reviewer cb1a,
>
> Thank you again for your suggestions. We have revised our paper to refer the reader to Appendix A of DuSell & Chiang (2020) for more details about the Hardest CFL, and we have included an explanation of our hyperparameter choices in the caption for Table 1.

---

### Decision · Program_Chairs · 2022-01-20

**Decision:**

Accept (Spotlight)

**Comment:**

This paper advances the long running thread of sequence modelling research focussed on differentiable instantiations of stack based models. In particular it builds upon recent work on the Nondeterministic Stack RNN (NS-RNN) by introducing three extensions. The first is to relax the need for a normalised distribution over the state and action distribution and allow unnormalised weights, this mostly serves to facilitate gradient flow and thus easier training. The second extension allows the RNN to condition on the top stack state as well as the symbol, improving expressiveness. The third improvement introduces a method for limiting the memory required to run the proposed model on long sequences, thus allowing its application to practical language modelling tasks. Each of these requires substantial algorithmic innovations.

The reviewers all agree that this is a strong paper worthy of publication. The paper includes a useful review of previous differentiable stack models which nicely sets up the rest of the paper where the contributions are well motivated and clearly presented. The reviewers had a number of clarification questions, partly due to the author's use of overly concise citations for key algorithms rather than inline descriptions. This situation has been improved by updates made to the paper.
The evaluation includes a series of synthetic experiments which are clear and provide a good elucidation of the various stack models properties. The practical evaluation on language modelling is more limited and serves mostly to demonstrate that the nondeterministic model can be scaled to a basic language modelling task.

Overall this is a strong paper with a well motivated and clear hypothesis. It provides a substantial extension to the prior work on nondeterministic stack models and progresses this line of research toward practical applications.